# Symbol tuning improves in-context learning in language models

**Jerry Wei**[1,2,*]     **Le Hou**[1]     **Andrew Lampinen**[1]     **Xiangning Chen**[1,*]     **Da Huang**[1]

**Yi Tay**[1]   **Xinyun Chen**[1]   **Yifeng Lu**[1]   **Denny Zhou**[1]   **Tengyu Ma**[1,2,†]   **Quoc V. Le**[1]

[1] Google      [2] Stanford University

## Abstract

We present *symbol tuning*—finetuning language models on in-context input–label pairs where natural language labels (e.g., "positive/negative sentiment") are replaced with arbitrary symbols (e.g., "foo/bar"). Symbol tuning leverages the intuition that when a model cannot use instructions or natural language labels to figure out a task, it must instead do so by learning the input–label mappings.

We experiment with symbol tuning across PaLM models up to 540B parameters and observe benefits across various settings. First, symbol tuning boosts performance on unseen in-context learning tasks and is much more robust to underspecified prompts, such as those without instructions or without natural language labels. Second, symbol-tuned models are much stronger at algorithmic reasoning tasks, with up to 18.2% better performance on the List Functions benchmark and up to 15.3% better performance on the Simple Turing Concepts benchmark. Finally, symbol-tuned models show large improvements in following flipped-labels presented in-context, meaning that they are more capable of using in-context information to override prior knowledge.

## 1 Introduction

A key feature of human intelligence is that humans can learn to perform new tasks by reasoning using only a few examples. Scaling up language models has unlocked a range of new applications and paradigms in machine learning, including the ability to perform challenging reasoning tasks via few-shot examples given in-context (Brown et al., 2020; Chowdhery et al., 2022; OpenAI, 2023, *inter alia*). Language models, however, are still sensitive to the way that prompts are given, indicating that they are not reasoning in a robust manner. For instance, language models often require heavy prompt engineering (Brown et al., 2020; Reynolds and McDonell, 2021) or phrasing tasks as instructions (Wei et al., 2022a; Ouyang et al., 2022; Sanh et al., 2022, *inter alia*), and they exhibit unexpected behaviors such as performance on tasks being unaffected even when shown exemplars with random labels (Min et al., 2022b) or flipped labels (Wei et al., 2023).

In this paper, we propose a simple finetuning procedure that we call *symbol tuning*, which significantly improves the ability of language models to reason with and learn from input–label mappings presented in-context. In the symbol-tuning procedure, we finetune language models on input–label pairs presented in-context where natural language labels are remapped to arbitrary symbols.[1] The intuition is that when models cannot rely on instructions or relevant natural language labels to figure out a given task, they must instead do so by reasoning with input–label mappings presented in-context in order to learn the mappings that reveal the task. We perform symbol tuning using a mixture of 22 NLP datasets with various arbitrary symbols as labels and experiment using instruction-tuned PaLM models (Flan-PaLM) with several sizes (8B, 62B, 62B-cont, 540B).

First, symbol tuning improves performance of baseline models on unseen in-context learning tasks across various settings (with/without instructions, with/without relevant labels), with larger performance gains when instructions or natural language labels are not given in the prompt. For example, when prompts do not contain instructions or relevant labels, symbol tuning yields a +11.1% average performance improvement across eleven evaluation tasks for Flan-PaLM-62B-cont.

---

[*] Work done as a Student Researcher at Google.

[†] Work done as a Visiting Researcher at Google.

[1] We call our method *symbol* tuning because arbitrary designation is a key property of symbols (Newell and Simon, 1976) and using symbols is a crucial part of intelligence (Newell, 1980; Santoro et al., 2021).

<table>
<tr><th>Instruction Tuning</th><th>Symbol Tuning</th></tr>
<tr><td>In-context exemplars not needed to learn the task</td><td>Must use in-context exemplars to learn the task</td></tr>
</table>

**Instruction Tuning**
In-context exemplars not needed to learn the task

**Symbol Tuning**
Must use in-context exemplars to learn the task

**Input**

*Instruction* — What is the sentiment of this?
*Exemplar* — *This movie is great*
*Label* — **Answer**: Positive  [Relevant]

*Instruction* — What is the sentiment of this?
*Exemplar* — *Worst film I've ever seen*
*Label* — **Answer**: Negative  [Relevant]

[more exemplars]

*Evaluation Example* — What is the sentiment of this?
*This movie is terrible*
**Answer**:

**Output**
Negative

**Input**

[None]
*This movie is great*
**Answer**: Foo  [Unrelated]

[None]
*Worst film I've ever seen*
**Answer**: Bar  [Unrelated]

[more exemplars]

[None]
*This movie is terrible*
**Answer**:

**Output**
Bar

Figure 1: We tune models on tasks where natural language labels are replaced with arbitrary symbols (*symbol tuning*). Symbol tuning relies on the intuition that when instructions and relevant labels are not available, models must use in-context exemplars to learn the task.

Second, symbol-tuned models are better at algorithmic reasoning tasks, a striking result since symbol tuning only includes natural language data and did not have any numerical or algorithmic data. On a set of reasoning evaluation suites for list functions (e.g., remove the last element in a list), symbol-tuned models experience performance improvements of **+18.2%** for Flan-PaLM-8B, **+11.1%** for Flan-PaLM-62B, and **+3.6%** for Flan-PaLM-540B. On a set of turing concept tasks (e.g., swapping 0s and 1s in a string), symbol-tuned models also improve by **+15.3%** for Flan-PaLM-8B and Flan-PaLM-62B and **+4.7%** for Flan-PaLM-540B.

Finally, we experiment on an in-context learning setting where inputs have flipped labels, which forces the model to override its prior knowledge when presented with contradictory information in-context. Pretrained language models have the ability to somewhat follow flipped labels—this ability is lost during instruction tuning but can be restored via symbol tuning. Overall, we hope that the strong empirical results from symbol tuning encourage further work in allowing language models to reason over arbitrary symbols given in-context.

## 2 Symbol tuning

Despite their ability to perform some reasoning tasks after being shown in-context exemplars (Chowdhery et al., 2022; OpenAI, 2023), language models are still sensitive to the way in which these tasks are presented in prompts (Brown et al., 2020;

Reynolds and McDonell, 2021; Wei et al., 2022a), suggesting that they are not reasoning in a robust way. Instruction tuning has been shown to improve performance and allow models to better follow in-context exemplars (Mishra et al., 2022; Min et al., 2022a; Wei et al., 2022a; Ye et al., 2021; Chung et al., 2022). One shortcoming, however, is that models are not forced to learn to use the exemplars because the task is redundantly defined in the evaluation example via instructions and natural language labels. For example, in the left-hand side of Figure 1, although the exemplars can help the model understand the task, they are not strictly necessary since the model could ignore the exemplars and just read the instruction.

To make the model better at in-context learning, we propose symbol tuning, in which the model is finetuned on exemplars where the instructions are removed and natural language labels are replaced with semantically-unrelated labels (e.g., "Foo," "Bar," etc.). In this setup, the task is unclear without looking at the in-context exemplars. For example, if the prompt from the previous paragraph was changed to "*<sentence>. Answer: {Foo, Bar}*" (as shown in the right-hand side of Figure 1), multiple in-context exemplars would be needed in order to figure out the task. Because symbol tuning teaches the model to reason over in-context exemplars, symbol-tuned models should have better performance on unseen tasks that require reasoning between in-context exemplars and their labels.

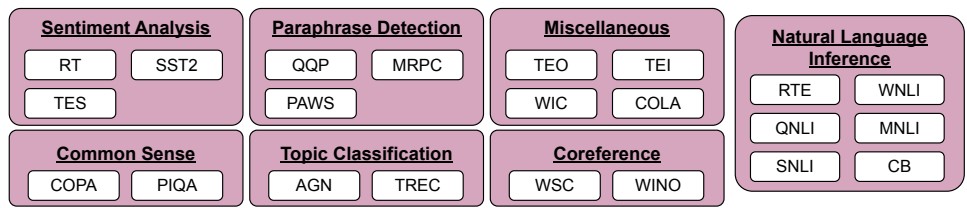

Figure 2: Datasets and task types used for symbol tuning. See Appendix D.1 for dataset details.

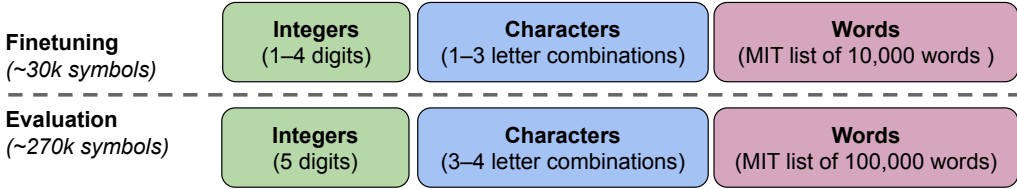

Figure 3: We use a set of ~300k arbitrary symbols from three categories (integers, character combinations, and words). ~30k symbols are used during tuning and the rest are held out for evaluation. See Appendix E.1 for more details on the symbols that we used.

## 3 Experimental setup

### 3.1 Tuning tasks & prompt formatting

Figure 2 shows the 22 publicly-available NLP datasets from HuggingFace (Lhoest et al., 2021) (see Appendix D.1 for dataset details) that we use for our symbol-tuning procedure (we ablate the number of datasets used for symbol tuning in Appendix B.4). We selected NLP tasks that have been widely used in the literature (Wang et al., 2018, 2019). Each dataset is categorized into one of seven task types—we only selected classification-type tasks because symbol tuning requires discrete labels. For each dataset, we use examples from the training split to compose prompts that we use for tuning. Each prompt uses a randomly-selected input–label format (formats are shown in Appendix E.2) and contains a randomly-selected number between 2 and 10 of in-context exemplars per class. We remap labels to a randomly-selected label from a set of ~30k labels from three label types as shown in Figure 3 (we ablate the number of labels in Appendix B.5 and the label types in Appendix B.6). Examples of generated tuning prompts for each task are shown in Appendix G.1. Code for generating arbitrary symbols can be found at https://github.com/JerryWeiAI/symbol-tuning.

### 3.2 Evaluation tasks

We want to evaluate a model's ability to perform on unseen tasks, so we cannot evaluate on tasks used in symbol tuning (22 datasets) or used during instruction tuning (1.8k tasks). Hence, we choose 11 NLP datasets from HuggingFace (Lhoest et al., 2021) that were not used in either stage of fine-tuning (details are shown in Appendix D.2): (Conneau and Kiela, 2018, **SUBJ**); (Basile et al., 2019, **TEH**); (Mohammad et al., 2016, **TEAB**); (Mohammad et al., 2016, **TEAT**); (Mohammad et al., 2016, **TEFE**); (Mohammad et al., 2016, **TEHI**); (Alex et al., 2021, **ADEC**); (Alex et al., 2021, **OR**); (Alex et al., 2021, **SOT**); (Alex et al., 2021, **TOS**); and (Alex et al., 2021, **TC**). We use the validation split of each dataset to generate evaluation prompts. For each dataset, we randomly select a maximum of 100 examples to use during evaluation. Each evaluation prompt uses a randomly-selected input–label format following Section 3.1, though we fix the number of in-context exemplars per class at $k = 4$ (we ablate this parameter in Appendix C.4).

We generate prompts for the four different in-context learning (ICL) settings described in Figure 4; each setting either contains or does not contain instructions describing the task (see Appendix D.2 for the instructions we use for each task) and does or does not contain relevant natural language labels. For settings that do not use relevant natural language labels, we remap original labels to a randomly-selected label from a set of ~270k semantically-unrelated labels as shown in Figure 3 (we removed labels that were seen during symbol tuning). Examples of generated evaluation prompts for each task are shown in Appendix G.2.

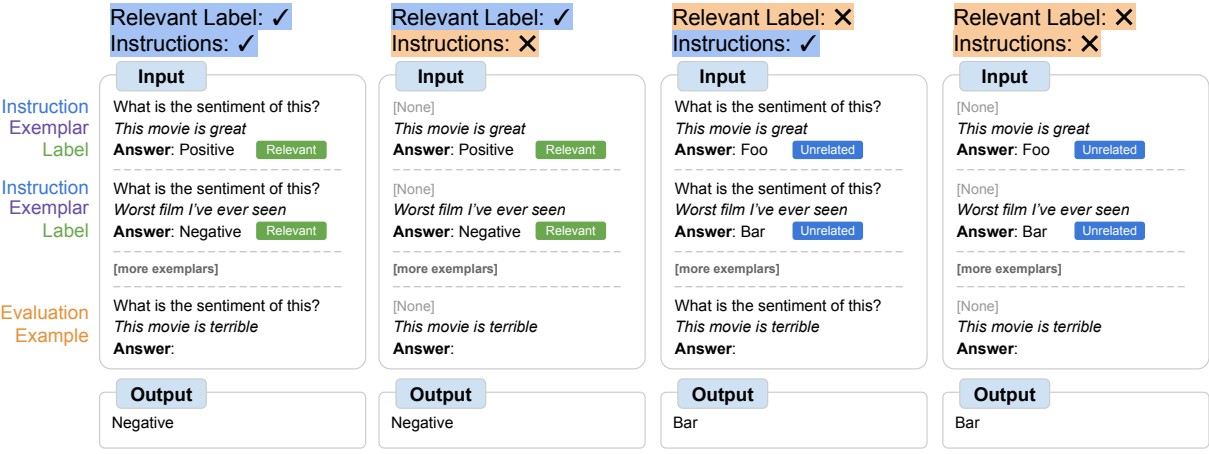

Figure 4: Depending on the availability of instructions and relevant natural language labels, models may need to do varying amounts of reasoning with in-context exemplars. When these features are not available, models must reason with the given in-context exemplars in order to successfully perform the task. When they are available, reasoning with exemplars can help but is not necessary.

## 3.3 Models & finetuning procedure

For our experiments, we tune Flan-PaLM, the instruction-tuned variants of PaLM. We use the instruction-tuned variants in order to reduce the number of steps needed for tuning, since symbol tuning an instruction-tuned model does not require relearning the information learned during the original round of instruction tuning. We use three different sizes of Flan-PaLM models: Flan-PaLM-8B, Flan-PaLM-62B, and Flan-PaLM-540B. We also tested Flan-PaLM-62B-cont (PaLM-62B at 1.3T tokens instead of 780B tokens); we abbreviate this model size as 62B-c.

Our symbol-tuning pipeline mixes all datasets and randomly samples from each dataset. To ensure that the dataset sizes are balanced (i.e., no dataset gets completely overshadowed), we limit the number of training examples per dataset to a maximum of 25k randomly-selected examples. Training examples are combined into a single sequence using packing (Raffel et al., 2020), and inputs are separated from labels using an end-of-sequence (EOS) token. We tune all models using a batch size of 32 and the Adafactor optimizer (Shazeer and Stern, 2018). For 8B and 62B models, we tune with a learning rate of $3 \times 10^{-3}$, and we tune Flan-PaLM-540B with a learning rate of $1 \times 10^{-3}$. We use 2048 and 512, respectively, as the input and target sequence lengths during tuning.

For 8B and 62B model evaluations, we report results from the checkpoint after tuning for 4k steps,

and for 540B model evaluations, we report results from the checkpoint after tuning for 1k steps (we ablate the number of tuning steps in Appendix B.2). See Appendix E.3 for the number of finetuning steps, learning rate, batch size, and dropout used for each model. As a baseline, we compare symbol-tuned models against Flan-PaLM models, and we also compare symbol tuning against continued instruction tuning in Appendix B.1.

## 4 Symbol-tuned models are better in-context learners

During symbol tuning, models must learn to reason with in-context exemplars in order to successfully perform tasks because prompts are modified to ensure that tasks cannot be learned from natural language labels or instructions. Symbol-tuned models should thus perform better in settings where tasks are unclear and require reasoning between in-context exemplars and their labels. Additionally, since symbol tuning is meant to improve the ability to follow in-context exemplars, it should not modify prior knowledge and should thus retain the same performance in settings where exemplars are not as necessary to complete the task.

To explore these settings, we define four ICL settings that vary the amount of reasoning required between inputs and labels in order to learn the task (based on the availability of instructions/relevant labels), as shown in Figure 4. The easiest of these settings uses prompts where both instructions and

| | Average performance on eleven tasks | | | |
|---|---|---|---|---|
| **Relevant labels:** | ✓ | ✓ | ✗ | ✗ |
| **Task instructions:** | ✓ | ✗ | ✓ | ✗ |
| Random Guessing | 42.4 | 42.4 | 42.4 | 42.4 |
| Flan-PaLM-8B | 63.9 | 61.6 | 42.4 | 44.2 |
| + Symbol tuning (ours) | 57.6 (**-6.3**) | 54.3 (**-7.3**) | 58.2 (**+15.8**) | 52.8 (**+8.6**) |
| Flan-PaLM-62B | 74.3 | 70.0 | 57.0 | 50.5 |
| + Symbol tuning (ours) | 75.5 (**+1.2**) | 70.8 (**+0.8**) | 71.4 (**+14.4**) | 60.3 (**+9.8**) |
| Flan-PaLM-62B-cont | 77.3 | 70.3 | 56.3 | 51.0 |
| + Symbol tuning (ours) | 78.9 (**+1.6**) | 74.5 (**+4.2**) | 71.8 (**+15.5**) | 62.1 (**+11.1**) |
| Flan-PaLM-540B | 82.2 | 77.4 | 70.7 | 58.1 |
| + Symbol tuning (ours) | 84.4 (**+2.2**) | 78.8 (**+1.4**) | 80.0 (**+9.3**) | 63.6 (**+5.5**) |

Table 1: Large-enough symbol-tuned models are better at in-context learning than baselines, especially in settings where relevant labels are not available. Performance is shown as average model accuracy (%) across eleven tasks (per-task results are shown in Appendix F.2).

relevant labels are available (as in-context exemplars are not necessary to learn the task), while the hardest setting uses prompts where instructions and relevant labels are both unavailable.

In Table 1, we evaluate model performance before and after symbol tuning in each of these settings. We find that symbol tuning improves performance across all ICL settings for models 62B and larger, with small improvements in settings with relevant natural language labels (+0.8% to +4.2%) and substantial improvements in settings without relevant natural language labels (+5.5% to +15.5%). Strikingly, when relevant labels are unavailable, symbol-tuned Flan-PaLM-8B outperforms Flan-PaLM-62B, and symbol-tuned Flan-PaLM-62B outperforms Flan-PaLM-540B. This performance difference suggests that symbol tuning can allow much smaller models to perform as well as large models on learning input-label mapping from exemplars (effectively saving ~10x inference compute).

Symbol-tuned models also perform somewhat-comparably in settings with only relevant labels or only instructions, unlike baseline models whose performance in settings with only relevant labels is always better than in settings with only instructions. Performance in settings with relevant labels actually decreases for Flan-PaLM-8B after symbol-tuning, however, which may suggest that symbol tuning a small model can override its prior knowl-

edge due to overfitting. Overall, the improvements demonstrate the strong potential of symbol tuning to improve model performance, especially when tasks require learning from in-context exemplars.

## 5 Symbol tuning improves algorithmic reasoning

Symbol tuning is designed to force the model to learn from input–label mappings in the in-context exemplars because the symbols are unrelated to the task and no instructions are provided (and thus the model cannot rely on any other guidance to determine the task). For this reason, we posit that symbol tuning should not only improve the model's ability to map natural language inputs to arbitrary symbols, but also its ability to learn other forms of input–label mappings such as algorithms.

To test this, we experiment on algorithmic reasoning tasks from BIG-Bench (Srivastava et al., 2022). We first experiment on a set of list function tasks (Rule et al., 2020; Srivastava et al., 2022) where the model needs to identify a transformation function (e.g., remove the last element in a list) between input and output lists containing non-negative integers. These tasks were evaluated in a four-shot setting, following our evaluation setup in Section 3.2. Additionally, we test models on a set of simple turing concepts (Telle et al., 2019; Srivastava et al., 2022) where models need to reason

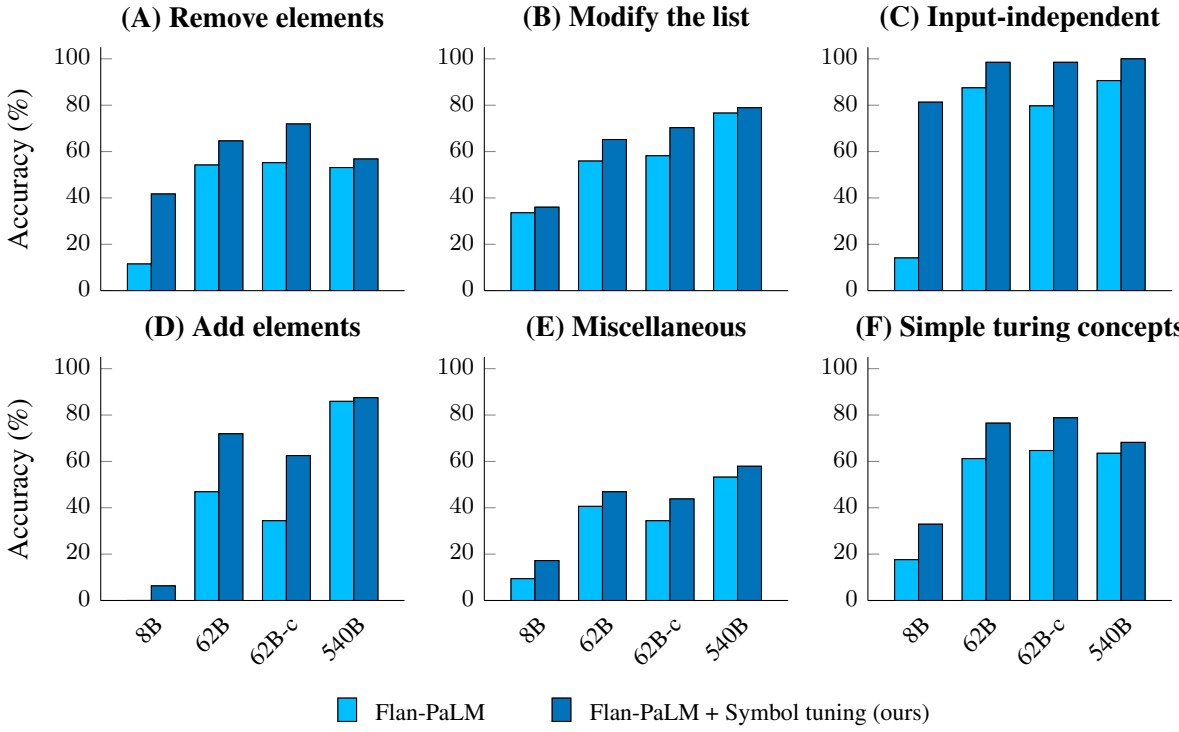

Figure 5: Symbol-tuned models achieve higher performance on list function tasks and simple turing concept tasks. (A–E): categories of list functions tasks (Rule et al., 2020; Srivastava et al., 2022). (F): simple turing concepts task (Telle et al., 2019; Srivastava et al., 2022). Accuracy per list function category is averaged across all subtasks (categories and per-task results are shown in Appendix F.1).

with binary strings to learn the concept that maps an input to an output (e.g., swapping 0s and 1s in a string). These tasks have predetermined shots for each evaluation example. We selected these algorithmic tasks because they test the model's ability to generalize to different task types (the symbol-tuning tasks were classification problems with discrete labels, while these tasks are more open-ended generation problems[2]) and do not require world knowledge (symbol tuning does not increase a model's prior knowledge).

In Figure 5, we show model performance on the twenty list function tasks with the highest human accuracy baselines[3] (Rule, 2020) separated into five categories (category details are described in Appendix F.1) and the turing concepts containing 3 or fewer instructions in the AS II subset of the simple turing concepts task. On the list function tasks, symbol tuning results in an average performance

improvement across all tasks of 18.2% for Flan-PaLM-8B, 11.1% for Flan-PaLM-62B, 15.5% for Flan-PaLM-62B-cont, and 3.6% for Flan-PaLM-540B. On the turing concept tasks, symbol tuning results in a performance improvement of 15.3% for Flan-PaLM-8B and Flan-PaLM-62B, 14.1% for Flan-PaLM-62B-cont, and 4.7% for Flan-PaLM-540B. Flan-PaLM-62B-cont with symbol tuning outperforms Flan-PaLM-540B on the list function tasks (in terms of average accuracy across tasks), which is equal to a ∼10x reduction in inference compute. These improvements on an unseen task type suggest that symbol tuning indeed strengthens the model's ability to learn in-context, as the symbol-tuning procedure did not have algorithmic data and only used natural language data.

## 6 Symbol-tuned models can override priors via flipped labels

Wei et al. (2023) showed that while pretrained language models (without instruction tuning) could, to some extent, follow flipped labels presented in-context, instruction tuning degraded this ability. Symbol tuning, on the other hand, forces models to

---

[2]Although chain-of-thought (Wei et al., 2022b, CoT) can be viewed as an open-ended generation problem, in Appendix C.2, we found that symbol-tuning did not significantly affect a model's CoT reasoning abilities, possibly because our symbol-tuning data did not incorporate any CoT prompts.

[3]We do not directly compare with the human baselines because our evaluation format was different.

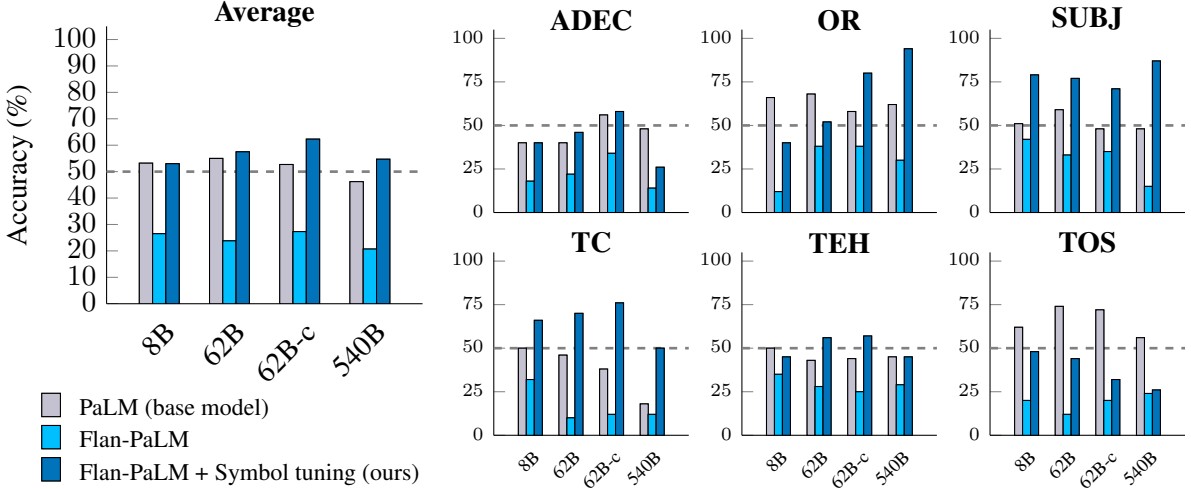

Figure 6: Symbol-tuned models are much better at following flipped labels presented in-context than instruction-tuned models are for all model sizes. Instruction-tuned models cannot flip predictions to follow flipped labels (performance is well below random guessing), while symbol-tuned models can do this more often (performance matches or is slightly above random guessing). Ground-truth labels for evaluation examples are flipped, so if a model learns to follow flipped labels, its accuracy should be above random guessing (e.g., a perfect model that can follow flipped labels should get 100% accuracy on our evaluations).

consider the label presented in-context as an arbitrary symbol, which should reduce the model's usage of prior knowledge that contradicts the flipped labels. For this reason, we expect that symbol tuning would be able to improve and restore the ability to follow flipped labels in-context.

To test this, we flip the labels of both in-context exemplars and the evaluation example for the tasks described in Section 3.2 (we remove tasks with more than two labels from this experiment since it is unclear how to best "flip" more than two labels). For example, for the SST2 dataset, all exemplars that are labeled as having "positive" sentiment will now be labeled as having "negative" sentiment. A perfect model that can follow these flipped labels should achieve 100% accuracy on these tasks if its accuracy in the standard ICL setting is also 100%.

As shown in Figure 6, symbol tuning restores the ability to follow flipped labels that was lost during instruction tuning. We see that there is a similar trend across all model sizes—instruction-tuned models are generally unable to follow flipped labels (as demonstrated by their performance being far below random guessing), but symbol-tuned models are much more capable of doing so. We found that after symbol tuning, Flan-PaLM-8B sees an average improvement across all datasets of 26.5%, Flan-PaLM-62B sees an improvement of 33.7%, and Flan-PaLM-540B sees an improve-

ment of 34.0%. For some datasets (e.g., OR, SUBJ, TC), symbol-tuned models can now override priors and follow flipped labels (i.e., achieve much better performance than random guessing), despite instruction-tuned models not being able to do so for any datasets. Additionally, symbol-tuned models match or beat pretraining-only models in terms of average performance, indicating that symbol tuning has, to some extent, restored the model's original ability to follow flipped labels.

These results further indicate another type of generalized in-context learning capability, as we did not include any flipped labels during symbol tuning. Although the performance improvement from symbol tuning is large, we note that more work should be done in this area since performance on the flipped-labels settings is, on average, not significantly better than random guessing.

## 7 Related work

### 7.1 In-context learning via semantic prior knowledge

Recent studies on in-context learning suggest that prior knowledge plays a significant role in how models learn in-context. For example, Wei et al. (2023) showed that some small models and instruction-tuned models cannot follow flipped labels presented in-context, suggesting that these models primarily utilize prior knowledge for in-

context learning. Min et al. (2022b) found a similar result that using random ground-truth labels in in-context exemplars does not significantly affect performance, meaning that performance may be driven by other factors such as the label space.

Reynolds and McDonell (2021) also showed that cleverly-constructed prompts in a zero-shot setting could outperform prompts in a few-shot setting, implying that, for some tasks, models can achieve better performance by leveraging their existing knowledge than from attempting to learn the task from in-context exemplars. Additionally, in chain-of-thought prompting (Wei et al., 2022b), Madaan and Yazdanbakhsh (2022) and Wang et al. (2022) showed that performance on multi-step reasoning tasks does not decrease when models are provided with logically-incorrect prompts. Raghu et al. (2020) also demonstrated that systems such as MAML can effectively "memorize" labels when trained in a way where all labels can be memorized, which further illustrates that, when possible, models may attempt to use prior knowledge rather than adapt to each new task.

Our findings do not dispute the idea that semantic prior knowledge can provide significant benefits to in-context learning. Indeed, we showed that instruction-tuned models cannot follow flipped labels in-context, which is consistent with the findings from Wei et al. (2023). We instead aim to demonstrate that through symbol tuning, language models can retain the benefits of utilizing prior knowledge while also improving their ability to learn from input–label pairs shown in-context.

## 7.2 In-context learning via in-context exemplars

At the same time, however, other recent work has suggested that language models can, in fact, learn in-context using the given exemplars. This ability may be more useful than the ability to use semantic prior knowledge because it would allow models to perform tasks that are not seen in or contradict pretraining data. Garg et al. (2022), for instance, showed that transformers trained from scratch can perform in-context learning on linear-regression tasks at a similar performance level as the least-squares estimator. This capability was shown to result from transformers implementing standard learning algorithms such as gradient descent (Akyürek et al., 2023; von Oswald et al.,

2022; Dai et al., 2023). Furthermore, Webson and Pavlick (2022) demonstrated that, in a natural language setting, language models can learn at the same rate during finetuning even when given irrelevant or misleading prompts. On a broader level, Rajendran et al. (2020) and Yin et al. (2020) found that adding noise to, shuffling, or regularizing the label space can make systems better at learning and adapting to new tasks.

In this paper, we attempt to improve the degree to which language models are able to learn tasks via input–label mappings. Our symbol-tuning method can be seen as a form of label augmentation and is thus similar to the proposed methods from Rajendran et al. (2020) and Yin et al. (2020), though it differs crucially in that we apply them to tune large language models. Additionally, We found that symbol-tuned models saw significant improvements in their ability to learn in-context (e.g., on algorithmic tasks or settings with underspecified prompts), which supports the idea that langauge models have the ability to learn in-context using the given exemplars.

## 7.3 Tuning language models

Our work presented symbol tuning, a form of finetuning on input–label pairs where labels are remapped to arbitrary symbols. Symbol tuning relates to a broader body of work showing that finetuning language models can significantly alter their behavior and performance in different settings. For example, Wei et al. (2022a) first presented instruction tuning (finetuning on tasks phrased as instructions) and showed that this finetuning procedure substantially improves model performance in zero-shot settings. Chung et al. (2022) further scaled this procedure by adding more tasks, increasing model sizes, and adding chain-of-thought data, demonstrating that, with these changes, tuned models are significantly better at chain-of-thought reasoning, open-ended generation, and several evaluation benchmarks.

Our experimental findings match these results in terms of showing that finetuning can significantly alter model performance. Our work differs, however, by not only focusing on settings with in-context exemplars and underspecified prompts, but also by modifying the finetuning procedure to make tasks harder to learn and require additional reasoning with in-context exemplars.

## 8   Limitations

While our study presents a simple yet effective method of improving in-context learning for language models, there are several limitations to our work. An open question is how to apply symbol tuning in a generative setting—we symbol tuned models on a range of classification tasks with discrete labels so that we can remap labels to arbitrary symbols, but we did not tune on generation tasks since it is unclear how to remap outputs to symbols in those settings. Future work could thus explore whether symbol tuning can be applied in a generative setting.

Additionally, our symbol-tuning procedure used 22 NLP datasets—while we ablated the number of datasets in Appendix B.4 and saw that increasing the number of datasets used for symbol tuning generally improves performance, we did not experiment with adding more tasks. Prior work, however, has demonstrated that scaling up finetuning methods can improve their impact on language models (Chung et al., 2022), so a natural extension would be to examine whether further scaling up the symbol-tuning method would have a similar result.

Furthermore, we applied symbol tuning to a family of instruction-tuned language models. It is unknown, however, whether the effects of symbol tuning that we showed may be affected by changes to the pretraining objective, model architecture, or training process. Similarly, symbol tuning may have different effects on language models that are not instruction tuned, as we did not specifically experiment on this factor. For this reason, future work may investigate how these factors impact the effectiveness of symbol tuning for improving in-context learning abilities in language models.

Because we only experimented with one family of language models, it is still unclear whether symbol tuning is effective for other models. Applying symbol tuning to other language models would likely require adjustments to the finetuning procedure to be successful (e.g., number of finetuning steps, mixing with previous data, number datasets), but a definitive conclusion about these factors cannot be drawn without further experimentation. We thus note that a crucial direction for future work is to explore how well symbol tuning translates to other language models.

## 9   Conclusions

In this paper, we presented *symbol tuning*, a new method of tuning models on tasks where natural language labels are remapped to arbitrary symbols. Symbol tuning is based off of the intuition that when models cannot use instructions or relevant labels to determine a presented task, it must do so by instead learning from in-context exemplars. We tuned four language models (Flan-PaLM-8B, Flan-PaLM-62B, Flan-PaLM-62B-cont, and Flan-PaLM-540B) using our symbol-tuning procedure, utilizing a tuning mixture of 22 datasets and approximately 30k arbitrary symbols as labels.

Experimentally, we showed that symbol tuning can significantly improve a model's ability to learn from in-context exemplars in not only natural language settings, but also on algorithmic tasks. First, we showed that symbol tuning improves performance on unseen in-context learning tasks, especially when prompts do not contain instructions or relevant labels. We also found that symbol-tuned models were much better at algorithmic reasoning tasks, despite the lack of numerical or algorithmic data in the symbol-tuning procedure. Finally, in an in-context learning setting where inputs have flipped labels, symbol tuning (for some datasets) reunlocks the ability to follow flipped labels that was lost during instruction tuning.

Through symbol tuning, we aim to have increased the degree to which models can examine and learn from input–label mappings during in-context learning. We hope that our results encourage further work towards improving language models' ability to reason over symbols presented in-context.

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
