# OpenReview forum: "Symbol tuning improves in-context learning in language models"
_EMNLP/2023/Conference — EMNLP 2023 Main_

### Official Review · Reviewer_MTeF · 2023-07-31

**Typos Grammar Style And Presentation Improvements:** The font does not seem to follow EMNL…
**Soundness:** 4

**Excitement:**

4: Strong: This paper deepens the understanding of some phenomenon or lowers the barriers to an existing research direction.

**Missing References:**

None

**Paper Topic And Main Contributions:**

This paper proposes symbolic tuning for in-context learning.
In symbol tuning, we replace the semantic labels of the few-shot exemplars with random irrelevant labels and remvoe the instructions in the exemplars. Then the instruction-tuned models are finetuned with the new data.
According to the authors, symbolic tuning not only improves performance on few-shot classification problems, but also enhances algorithmic reasoning ability and can restore the flipped-label following ability.

**Questions For The Authors:**

1) What is the performance of symbol tuning an LLM without instruction tuning? Will it still increase its ability on algorithmic reasoning and following flipped labels?
2) What is the relationship betweem instruction tuning and symbol tuning (only for classification problems)?
3) Does symbol tuning increase algorithmic reasoning in CoT settings, like 8-shot CoT on GSM8K, which is also open-ended generation like list function tasks?

**Reasons To Accept:**

1. The symbol tuning approach is simple and effective. It seems to be complementary to instruction tuning in terms of the algorithmic reasoning ability and strict instruction-following ability (e.g. flipped labels).
2. The extensive experiments across model scales make the results convincing.
3. The details of the experiments are available.

**Reasons To Reject:**

1. The authors only experiment with one class of closed-source models which are inaccessible to most researchers. The effectiveness of symbol tuning on the open-source LLMs such as Llama 1,2, is till unclear.
2. The template does not look like EMNLP 2023 template.

**Reproducibility:**

3: Could reproduce the results with some difficulty. The settings of parameters are underspecified or subjectively determined; the training/evaluation data are not widely available.

**Reviewer Confidence:**

4: Quite sure. I tried to check the important points carefully. It's unlikely, though conceivable, that I missed something that should affect my ratings.

---

> ### Author Rebuttal · Authors · 2023-08-25
>
> Thank you for the encouraging comments and feedback. Our paper explored a new method of finetuning which can improve algorithmic reasoning and in-context learning, and we are glad that you found our approach to be simple, effective, and detailed.
>
> We revised the manuscript based on your feedback and believe it is more clear (we cannot revise the submission during the review process but will do so as soon as possible). Please let us know if you have any further comments or suggestions.
>
> > What is the performance of symbol tuning an LLM without instruction tuning? Will it still increase its ability on algorithmic reasoning and following flipped labels?
>
> Thanks for this question about applying symbol tuning to non-instruction-tuned language models. In our paper, we applied symbol tuning to the instruction-tuned variants of language models since symbol tuning an instruction-tuned model means that the model does not need to relearn the information learned during the instruction-tuning process. We agree, however, that an open question is whether symbol tuning would be affected by whether the language model has undergone instruction tuning, as we did not experiment with symbol tuning base models. As mentioned in our limitations section, a limitation of our work is that we only experimented using a family of instruction-tuned internal language models, so it is unclear if symbol tuning would be affected by various changes in the language model.
>
> To make this limitation more clear, we modified Section 8 as follows:
>
> “Furthermore, we experimented with symbol tuning using a family of internal language models and their instruction-tuned variants. It is unknown, however, whether the effects of symbol tuning that we showed may be affected by changes to the pretraining objective, model architecture, or training process. For this reason, future work may investigate how these factors impact the effectiveness of symbol tuning for improving in-context learning abilities in language models.” → “Furthermore, we applied symbol tuning to a family of instruction-tuned internal language models. It is unknown, however, whether the effects of symbol tuning that we showed may be affected by changes to the pretraining objective, model architecture, or training process. Similarly, symbol tuning may have different effects on language models that are not instruction tuned, as we did not specifically experiment on this factor. For this reason, future work may investigate how these factors impact the effectiveness of symbol tuning for improving in-context learning abilities in language models.”
>
> > What is the relationship between instruction tuning and symbol tuning (only for classification problems)?
>
> Thanks for this comment about how symbol tuning is related to instruction tuning. In Figure 1, we provided a comparison of instruction tuning and symbol tuning. These two finetuning procedures are related in the sense that they both show the model some in-context exemplars for a given NLP task and then provide an evaluation example.
>
> The key difference between them, however, is that instruction tuning specifies the task with instructions and relevant natural language labels. Thus, in instruction-tuning prompts, models do not necessarily need to look at the in-context exemplars to learn the task, as they could instead just read the instruction, for example. In symbol tuning, on the other hand, we underspecify the task by removing instructions and providing irrelevant labels. This means that the model can only learn the task by reasoning over in-context exemplars to learn the relationship between inputs and labels.
>
> Please let us know if this helps clarify the relationship between instruction tuning and symbol tuning.
>
> > Does symbol tuning increase algorithmic reasoning in CoT settings, like 8-shot CoT on GSM8K, which is also open-ended generation like list function tasks?
>
> Thanks for allowing us to elaborate on how symbol tuning might improve algorithm chain-of-thought (CoT) reasoning. We also were interested in how symbol tuning might translate to CoT settings, and in Appendix B.2, we experimented on whether symbol-tuned models were better at MMLU and BIG-Bench-Hard, two well-known benchmarks.
>
> We found that there was no significant improvement on these benchmarks in a CoT setting (see Figure 14), despite the fact that the intermediate steps in CoT could be viewed as a symbolic mapping problem. This result may actually be expected, however, as our symbol-tuning data did not include any data with CoT reasoning (instead only containing input–label mappings without any intermediate reasoning steps). Although we did not explore how to apply symbol tuning to CoT data in this work because it is unclear how to best replace intermediate steps with symbols, we agree that the idea is an exciting direction that future work can explore.
>
> To make this section more noticeable for readers, we added a footnote in Section 5 following “while these tasks are more open-ended generation problems”:
>
> “Although chain-of-thought (Wei et al., 2022b, CoT) can be viewed as an open-ended generation problem, in Appendix C.2, we found that symbol-tuning did not significantly affect a model’s CoT reasoning abilities, possibly because our symbol-tuning data did not incorporate any CoT prompts.”
>
> > The authors only experiment with one class of closed-source models [...] the effectiveness of symbol tuning on the open-source LLMs such as Llama 1,2, is still unclear.
>
> Thank you for bringing up this important comment about the generalizability of symbol tuning towards other families of language models. We agree that this is a limitation of our work, as successfully applying symbol tuning to other language models would likely require some modifications that are currently unknown. For this reason, we added a paragraph discussion this limitation in Section 8:
>
> “Because we only experimented with one family of language models, it is still unclear whether symbol tuning is effective for other models. Applying symbol tuning to other language models would likely require adjustments to the finetuning procedure to be successful (e.g., number of finetuning steps, mixing with previous data, number datasets), but a definitive conclusion about these factors cannot be drawn without further experimentation. We thus note that a crucial direction for future work is to explore how well symbol tuning translates to other language models.“
>
> > The font does not seem to follow the EMNLP2023 template.
>
> Thank you for pointing out this detail, we have fixed this in the revised manuscript. With this corrected font, our manuscript is still within the page limits after moving the limitations section to be in the extra space allowed after the 8th page (as allowed by the EMNLP style, 2023.emnlp.org/calls/style-and-formatting/). Our revised manuscript with the font change conforms to the 9-page limit allowed for accounting for reviewer comments (2023.emnlp.org/calls/main_conference_papers/#long-papers).

---

### Official Review · Reviewer_F3QV · 2023-08-07

**Soundness:** 4

**Excitement:**

4: Strong: This paper deepens the understanding of some phenomenon or lowers the barriers to an existing research direction.

**Paper Topic And Main Contributions:**

What is this paper about:
This paper proposes a new method called symbol tuning to overcome the disadvantages of instruction tuning. In symbol tuning, the instructions are removed and natural language outputs are replaced with a random symbol. The LLMs are the fine-tuned on the input-masked outputs. This forces the models to reason only from examples not from the instructions. Experimental results on several NLP tasks shows the effectiveness of the proposed approach.

Main Contributions:
1. New method symbol tuning to effectively learn from in-context examples.
2. Proposed approach helps model to override knowledge if provided with contradictory information aka. editing knowledge in LLMs.
3. Strong results on all benchmarks compared to IT.

**Questions For The Authors:**

1. Will the authors make their code once accepted?
2. Why does symbol tuning performs worse for smaller LMs like Flan-PaLM-8B?
3. In Table-1 we see for the IT-LLM-62B-cont that without instructions performed better than with instructions? So, in this case even without instructions the model was able to learn from in-context examples. Does it mean we don't need symbol tuning in all the cases?

**Reasons To Accept:**

Reasons to Accept:
Strong motivation: The sensitiveness of prompts in the LLMs performance has been a concern in IT. This paper provided a strong motivation behind introducing symbol tuning.

Usefulness to the NLP community: The symbol tuning approach is useful to NLP community for training LLMs using in-context examples.

Strong experiment results: The authors performed comprehensive experiments on a broad range of NLP tasks and with a varied sizes of LLMs.

**Reasons To Reject:**

None.

**Reproducibility:**

3: Could reproduce the results with some difficulty. The settings of parameters are underspecified or subjectively determined; the training/evaluation data are not widely available.

**Reviewer Confidence:**

3: Pretty sure, but there's a chance I missed something. Although I have a good feel for this area in general, I did not carefully check the paper's details, e.g., the math, experimental design, or novelty.

---

> ### Author Rebuttal · Authors · 2023-08-25
>
> Thank you for the detailed comments and important questions. We presented symbol tuning, a new finetuning method that helps models learn in-context by forcing them to use in-context exemplars to learn a task. We are happy that you found the method to be effective and our results to be strong, useful, and comprehensive.
>
> Based on your feedback, we have added revisions to the manuscript (we cannot revise the submission during the review process but will do so as soon as possible)—please feel free to let us know if you have additional comments or questions.
>
> > Will the authors make their code [available] once accepted?
>
> Thanks for this nice suggestion to make code available. While we are unable to release our entire pipeline for generating symbol-tuning data, we released the script for generating arbitrary symbols, as the use of symbols as labels is the primary contribution of our paper (the rest of the pipeline includes collecting datasets from HuggingFace, replacing labels with the generated symbols, and placing input–label pairs into prompts). Note that we provide example prompts and implementation details in the Appendix so that generating symbol-tuning data should be a straightforward implementation.
>
> The anonymized repository can be found here:
>
> https://anonymous.4open.science/r/symbol-tuning-2FD5/
>
> We also added a mention of this code in Section 3 (we will replace the anonymized repository with the non-anonymized version after the reviewing process):
>
> “Code for generating arbitrary symbols can be found at https://anonymous.4open.science/r/symbol-tuning-2FD5/.”
>
> > Why does symbol tuning perform worse for smaller LMs like [IT-LLM-8B]?
>
> Thank you for this comment on how the effects of symbol tuning can vary with respect to model size. While symbol tuning improves performance for both small and large language models on algorithmic reasoning and flipped label settings, in Section 4 we indeed found that symbol tuning may hurt performance when relevant natural language labels are available. In this setting, symbol tuning a large-enough model still improves performance, but symbol tuning the smallest model (IT-LLM-8B) resulted in a performance drop.
>
> In our manuscript, we hypothesized that this could have been a result of the smallest model overfitting to the symbol-tuning data. This hypothesis is supported by the finding that holding the availability of task instructions constant, the performance with and without relevant labels is very similar (e.g., with task instructions: no relevant labels = 58.2%, with relevant labels = 57.6%). This may suggest that the model does not draw any additional semantic information from the relevant natural language labels that are included and may instead treat them as an arbitrary symbol, which could explain why performance with irrelevant labels matches performance with relevant labels.
>
> We added further discussion on this question in the “frequently asked questions” section in the Appendix:
>
> “In Section 4, we found that for our smallest model (IT-LLM-8B), symbol tuning may result in performance drops in in-context learning settings where relevant natural language labels are available. Our primary hypothesis for this behavior is that via the symbol-tuning procedure, the model learns to override its prior knowledge and overfit to the symbol-tuning data. This could result in the model learning to treat any label as an arbitrary symbol to learn, regardless of whether the label is relevant to the task or not. A key finding that supports this hypothesis is that the model achieves similar performance whether the labels are relevant to the task or not—as shown in Table 1, when task instructions are available, the model achieves 57.6% accuracy when the labels are relevant to the task and 58.2% accuracy when the labels are not relevant to the task, and when task instructions are not available, the model achieves 54.3% accuracy when the labels are relevant and 52.8% accuracy when the labels are not relevant. This finding implies that the model does not consider the semantics of the relevant labels and instead treats them just like any other arbitrary symbol, which could further suggest that the model has overfit to the symbol-tuning data by learning to treat *any* label as a symbol to learn.”
>
> > In Table-1 we see IT-LLM-62B-cont without instructions performed better than with instructions? [...] Does it mean we don't need symbol tuning in all cases?
>
> Thanks for this important comment on whether there are cases in which symbol tuning is not needed. In Table 1, we indeed found that the performance improvement from adding symbol tuning is much larger for settings where relevant labels or task instructions are not available.
>
> The most important factor is whether relevant natural language labels are available. In settings with relevant labels, symbol tuning improves performance by up to 4.2%, but in settings without relevant labels, symbol tuning improves performance by up to 15.5%. The large difference in improvements from symbol tuning suggests that symbol tuning is most effective when the task is underspecified. For tasks that are well-defined (e.g., instructions and relevant labels are both included), symbol tuning does not have a significant effect on performance (though it may still slightly improve performance by up to 2.2%).
>
> We have added discussion about this question in a new “frequently asked questions” section in the Appendix:
>
> “In Section 4, Section 5, and Section 6, we demonstrated that symbol tuning can significantly improve a language model’s in-context learning and algorithmic-reasoning abilities. At the same time, however, there are also some settings for which symbol tuning did not significantly improve performance. For example, in Section 4, we found that performance gains from symbol tuning are most significant when relevant natural language labels or task instructions are not available, and that symbol tuning results in smaller improvements when relevant labels are available. We also found that after symbol tuning, the smallest model (IT-LLM-8B) actually saw a decrease in performance when relevant labels are available, which may suggest that a small model overrides its prior knowledge and overfits to the symbol-tuning data. Moreover, we found that symbol tuning did not necessarily improve model performance on benchmarks such as MMLU and BIG-Bench-Hard (Appendix C.1), in chain-of-thought (Wei et al., 2022b) settings (Appendix C.2), and in zero shot settings (Appendix C.3).”

---

### Official Review · Reviewer_43HS · 2023-08-13

**Typos Grammar Style And Presentation Improvements:** ln. 60
**Soundness:** 5

**Excitement:**

4: Strong: This paper deepens the understanding of some phenomenon or lowers the barriers to an existing research direction.

**Paper Topic And Main Contributions:**

In this paper, the authors provide a novel finetuning approach for LLM via “symbol tuning”--finetuning LMs on input-label pairs for which the natural language labels are replaced with nonce words (“foo”,”bar”)--and conduct experimental validation of the proposed approach for in-context learning and algorithmic reasoning tasks. The authors find that symbol tuned models demonstrate superior performance on in-context learning tasks and with underspecified prompts. They further show that symbol tuning improves performance on multiple algorithmic reasoning benchmarks (BIG-Bench, Turing Concepts). The authors also conduct tests with prompts that have flipped labels, necessitating models’ overriding of prior knowledge to correctly answer prompts, showing that symbol tuning improves model performance in this domain as well.

**Reasons To Accept:**

* Novelty: the authors provide a novel approach to finetuning models, drawing from the intuition that when LMs struggle using instructions or labels to determine the task at hand, they instead resort to learning from in-context examples. This simple but effective approach provides both unique findings and a compelling explanation for model’s improvement with symbol tuning.
* Robustness: the authors support their methodology with clear and logical analysis on the impact of symbol tuning in a variety of different contexts–providing robust findings while also demonstrating the need for further research on the impact of symbol tuning for LLMs; this work provides promise for future work exploring more creative uses of input-label pairs during in-context learning.


**Reasons To Reject:**

Minor weaknesses/nits
* Reproducibility is an issue due to the models tested being closed-source (although the process of symbol-tuning is described in detail)
* Although the authors hypothesize about why symbol tuning allows models to perform better in the tested in-context learning and algorithmic reasoning tasks, the research lacks clear evidence backing these model’s performance (but this may be better kept for future work)


**Reproducibility:**

2: Would be hard pressed to reproduce the results. The contribution depends on data that are simply not available outside the author's institution or consortium; not enough details are provided.

**Reviewer Confidence:**

3: Pretty sure, but there's a chance I missed something. Although I have a good feel for this area in general, I did not carefully check the paper's details, e.g., the math, experimental design, or novelty.

---

> ### Author Rebuttal · Authors · 2023-08-25
>
> Thank you for the insightful comments and feedback. Our work proposes a simple finetuning procedure that can significantly improve a language model’s in-context learning and algorithmic reasoning abilities, and we are glad that you found our approach to be novel yet simple and effective.
>
> Please let us know if you have any further suggestions—we would be happy to continue working to improve our paper.
>
> > Reproducibility is an issue due to the models tested being closed-source
>
> Thanks for this important comment on the reproducibility of our results. Because our models were internal, we could not release our symbol-tuned model checkpoints, though we made sure to describe the symbol-tuning procedure in as much detail as possible, as you mentioned. Additionally, we made sure that all of our finetuning and evaluation datasets were publicly-available on HuggingFace.
>
> > Although the authors hypothesize about why symbol tuning allows models to perform better in the tested in-context learning and algorithmic reasoning tasks, the research lacks clear evidence backing these model’s performance
>
> Thank you for providing this discussion on why symbol tuning improves in-context learning. Our idea leverages the intuition that traditional instruction tuning overspecifies tasks and does not necessarily require models to even look at in-context exemplars, and we proposed symbol tuning as a way to force model to use in-context exemplars to learn the task by underspecifying the task via removing instructions and replacing labels with arbitrary symbols. Experiments in Appendix A.1, A.7, B.2, and B.3 are consistent with this. For even more evidence, experiments on analyzing specific datasets or applying symbol tuning in generative settings could shed further light, which we discussed in Section 8 (our limitations/future work section).

---

### Meta-Review · Area_Chair_Y9sD · 2023-09-25

**Recommendation:** 5

**Metareview:**

The paper proposes symbol tuning, a novel fine-tuning method for LLMs that uses random symbols instead of natural language labels and instructions. The paper shows that symbol tuning boosts LLMs’ performance on in-context learning and algorithmic reasoning tasks, and enables them to override prior knowledge with contradictory information. The paper tests symbol tuning on various benchmarks.
The reviewers like the paper for its novelty, effectiveness, motivation, analysis  and comprehensive experiments. The reviewers  only mention some minor issues, as well as reproducibility problem due to the closed-source models. The reviewers recommend accepting the paper, as it offers a simple yet effective  technique for improving LLMs’ in-context learning and algorithmic reasoning abilities. The paper also opens new possibilities for creative uses of input-label pairs during in-context learning.

Pros:
The symbol tuning approach is simple, effective, and seems to complement instruction tuning.
Novelty: The authors provide a novel and effective approach to fine-tuning models.
Robustness: The methodology is supported with clear and logical analysis on the impact of symbol tuning in various contexts.
Strong Motivation: The paper addresses the concern of prompt sensitivity in LLMs performance, providing a strong motivation for introducing symbol tuning.
Usefulness to the NLP Community: The symbol tuning approach is beneficial for training LLMs using in-context examples.
Strong Experiment Results: Comprehensive experiments were performed on a broad range of NLP tasks and with varied sizes of LLMs.
Detailed Experiments: The details of the experiments are available, making the results convincing.

Cons:

Reproducibility Issue: Due to the models tested being closed-source, reproducibility could be an issue, although the process of symbol-tuning is described in detail.
Limited Experimentation: The authors only experiment with one class of closed-source models which are inaccessible to most researchers. The effectiveness of symbol tuning on open-source LLMs such as Llama 1,2, is still unclear.
Lack of Evidence: While the authors hypothesize why symbol tuning allows models to perform better, there’s a lack of clear evidence backing these model’s performance.

Please note that a reviewer  mentioned template Issue: The template does not look like EMNLP 2023 template.

---

### Decision · Program_Chairs · 2023-10-07

**Decision:**

Accept-Main

**Comment:**

The paper proposes symbol tuning, a novel fine-tuning method for LLMs that uses random symbols instead of natural language labels and instructions. The paper shows that symbol tuning boosts LLMs’ performance on in-context learning and algorithmic reasoning tasks, and enables them to override prior knowledge with contradictory information. The paper tests symbol tuning on various benchmarks.
The reviewers like the paper for its novelty, effectiveness, motivation, analysis  and comprehensive experiments. The reviewers  only mention some minor issues, as well as reproducibility problem due to the closed-source models. The reviewers recommend accepting the paper, as it offers a simple yet effective  technique for improving LLMs’ in-context learning and algorithmic reasoning abilities. The paper also opens new possibilities for creative uses of input-label pairs during in-context learning.

Pros:
The symbol tuning approach is simple, effective, and seems to complement instruction tuning.
Novelty: The authors provide a novel and effective approach to fine-tuning models.
Robustness: The methodology is supported with clear and logical analysis on the impact of symbol tuning in various contexts.
Strong Motivation: The paper addresses the concern of prompt sensitivity in LLMs performance, providing a strong motivation for introducing symbol tuning.
Usefulness to the NLP Community: The symbol tuning approach is beneficial for training LLMs using in-context examples.
Strong Experiment Results: Comprehensive experiments were performed on a broad range of NLP tasks and with varied sizes of LLMs.
Detailed Experiments: The details of the experiments are available, making the results convincing.

Cons:

Reproducibility Issue: Due to the models tested being closed-source, reproducibility could be an issue, although the process of symbol-tuning is described in detail.
Limited Experimentation: The authors only experiment with one class of closed-source models which are inaccessible to most researchers. The effectiveness of symbol tuning on open-source LLMs such as Llama 1,2, is still unclear.
Lack of Evidence: While the authors hypothesize why symbol tuning allows models to perform better, there’s a lack of clear evidence backing these model’s performance.

Please note that a reviewer  mentioned template Issue: The template does not look like EMNLP 2023 template.